# sTREM-1, HMGB1, CRP, PCT, sCD14-ST, IL-6, IL-10, sHLA-G, and Vitamin D in Relation to Clinical Scores and Survival in SIRS/Sepsis

**DOI:** 10.3390/biomedicines13102481

**Published:** 2025-10-11

**Authors:** Michaela Kopcova, Anna Dobisova, Magda Suchankova, Elena Tibenska, Kinga Szaboova, Juraj Koutun, Maria Bucova

**Affiliations:** 1Institute of Immunology, Faculty of Medicine, Comenius University, 813 72 Bratislava, Slovakia; michaela.kopcova@fmed.uniba.sk (M.K.); magda.suchankova@fmed.uniba.sk (M.S.); 21st Department of Anesthesiology and Intensive Care Medicine, Faculty of Medicine, Comenius University and University Hospital, 821 01 Bratislava, Slovakia; anna.dobisova@fmed.uniba.sk (A.D.); juraj.koutun@ru.unb.sk (J.K.); 3Department of Immunology, Medirex Ltd., 820 16 Bratislava, Slovakia; elena.tibenska@medirex.sk (E.T.); kinga.szaboova@medirex.sk (K.S.)

**Keywords:** IL-10, presepsin, procalcitonin, sepsis survival, TREM-1, vitamin D

## Abstract

**Background:** Sepsis is a life-threatening organ dysfunction caused by a dysregulated host response to infection and remains a major cause of mortality in intensive care units. **Methods**: We analyzed plasma levels of sTREM-1, CRP, PCT, sCD14-ST, HMGB1, IL-6, IL-10, vitamin D (VD), and sHLA-G in patients with SIRS/sepsis, and assessed their relationships with APACHE II, SOFA scores, and survival. **Results:** Septic patients showed significantly elevated sTREM-1, CRP, PCT, sCD14-ST, and higher neutrophil-to-lymphocyte ratio, while VD levels were markedly reduced. Logistic regression identified CRP and PCT as the strongest univariate predictors of sepsis, but after adjustment for age, sex, BMI, and comorbidities, CRP lost significance, whereas VD and sCD14-ST remained independent predictors. Prognostically, higher IL-10 levels significantly correlated with 7- and 28-day mortality and with SOFA scores, while higher VD concentrations predicted better survival. **Conclusion:** CRP, PCT, and sCD14-ST are reliable diagnostic biomarkers of sepsis, with sTREM-1 providing additional value for disease monitoring. After adjustment for clinical covariates, VD emerged as an independent protective factor, whereas elevated IL-10 significantly predicted 7- and 28-day mortality. These findings underscore the utility of combining inflammatory and immunoregulatory biomarkers to improve sepsis diagnostics and prognostication, warranting validation in larger multicenter cohorts.

## 1. Introduction

Sepsis is defined as a life-threatening organ dysfunction caused by an individual’s dysregulated response to infection [1,2]. Sepsis represents a multifaceted response of the organism to a given infecting pathogen or its components, which can be amplified by endogenous factors. Disorders also occur in non-immunological systems, such as cardiovascular, hormonal, nervous, and others, leading to the development of difficult-to-regulate chaos. Septic shock as a subtype of sepsis is characterized by circulatory and cellular/metabolic dysfunction that is associated with an increased risk of death [3]. Despite the availability of modern therapeutic interventions, even in developed countries, sepsis is still one of the leading causes of hospitalization and early mortality of patients in intensive care units and later disability of patients having survived sepsis [4,5,6].

The immune response in sepsis is complex and dynamic. Immune system activation, hyperinflammation, and cytokine storm may lead to early multiple organ failure, whereas counter-regulatory mechanisms can induce immunosuppression, predisposing patients to secondary polymicrobial infections and late mortality. These overlapping phenomena, described as SIRS (systemic inflammatory response syndrome), CARS (compensatory anti-inflammatory response syndrome), and PICS (persistent inflammation immunosuppression and catabolism syndrome), which develops after surviving sepsis, illustrate the heterogeneous and evolving nature of the septic response [7,8,9,10].

The pitfalls of successful therapeutic management of sepsis are still early diagnosis and immediate initiation of targeted antibiotic treatment. The sensitivity and specificity of diagnostic methods, currently used as the gold standard, are still not sufficiently satisfactory.

In terms of sepsis development, its diagnostics and monitoring, TREM-1 (triggering receptor expressed on myelocytes) molecule has attracted attention as a potential contributor to hyperinflammation that might contribute to the dysregulated immune response and septic shock development [11]. The membrane bound TREM-1 is a pattern recognition receptor and plays a critical role in the immune response [12]. TREM-1 work together with Toll-Like Receptors (TLRs) to amplify inflammatory responses. TLRs recognize various microbial and endogenous ligands, and signaling then upregulates TREM-1 expression. After TLRs triggers an initial inflammatory cascade, activation of TREM-1 initiates results in the production of pro-inflammatory cytokines and chemokines, and functions as an amplifier of inflammation. Although, TREM-1 was initially predominantly associated with infectious diseases (first in response to bacterial lipopolysacharide), it plays a great role also in sterile inflammatory diseases [13,14,15,16,17].

Soluble sTREM-1 can be actively produced by e.g. monocytes and macrophages or passively released from necrotic cells, which occur in large numbers during sepsis. sTREM-1 levels rise significantly during infections, but an increase, though lower, occurs also during non-infectious inflammation [18].

During infectious and non-infectious inflammation besides TLR and TREM-1 pathways, so called STING (stimulator of interferon genes), pathway can be triggered. This pathway is activated by exogenous DNA, such as bacterial DNA, viral DNA and other exogenous DNA fragments, endogenous DNA, such as cytosol self-DNA and cytoplasmic chromatin fragment leakage. Many nucleic acid sensors have been identified to detect pathogen and cellular damage. It remains unclear whether STING is activated independently by septic stimuli or as a downstream effect of TLR engagement [19,20].

Another diagnostic and monitoring marker of sepsis, not commonly analyzed in septic patients, is the late pro-inflammatory cytokine and alarmin HMGB1 (high mobility group box 1). This intranuclear protein, which appears in the circulation 16–32 h after the activation of immunocompetent cells by endotoxin and interferon γ (IFN-γ), but its synthesis persists, which brings the possibility of expanding the diagnostic [21]. After being passively released from damaged and necrotic cells or actively produced by many cell types—mainly activated monocytes and macrophages, hepatocytes (in sepsis) to the extracellular milieu, it exerts potent pro-inflammatory activities [22,23,24].

Within our project, due to its immunomodulating properties and importance in anti-infectious immunity, we also focused our attention on vitamin D (VD) and investigated the relationship of its plasma concentrations to the course of the disease, clinical condition, laboratory findings and patient prognosis. VD deficiency is common in the general population and is also present in approximately 76% of critically ill patients [25]. The presence of VD deficiency in critically ill patients is also confirmed by the results of our previous work revealing reduced or insufficient levels of VD in the plasma of patients hospitalized at ICU for a systemic inflammatory response of infectious and non-infectious origin, while the lowest plasma concentrations of VD (severe deficiency—25(OH)D level < 12 ng/mL) were present in septic shock patients [26]. 

The aim of our study was to examine the diagnostic and prognostic potential of inflammatory biomarkers—sTREM-1, HMGB1 (high mobility group box protein), PCT (procalcitonin), presepsin (sCD14-ST), C-reactive protein (CRP), pro-inflammatory cytokine IL-6 (interleukin-6), immunoregulatory and anti-inflammatory cytokine IL-10, immunosuppressive and anti-inflammatory molecule sHLA-G (soluble human leukocyte antigen-G), and other molecules and cell parameters involved in immunity and inflammation. All these markers can contribute to the understanding of complex immunopathogenesis of sepsis and to outline the possibilities of using these molecules as prospective biomarkers in the early diagnosis, monitoring and prognosis of sepsis.

By analyzing the relationship of these biomarkers with clinical scores and outcomes, we sought to identify markers that may improve early diagnosis, monitoring, and prognosis.

Previous studies have mainly focused on individual biomarkers in sepsis, such as PCT, CRP, or sTREM-1, but few have assessed a broader immunological panel within the same patient cohort. Our study addresses this gap by simultaneously evaluating sTREM-1, HMGB1, sCD14-ST, IL-10, vitamin D, and sHLA-G in patients with SIRS and sepsis. This integrated approach enables direct comparison of their diagnostic and prognostic potential in a uniform clinical setting. By combining inflammatory and regulatory biomarkers and their association with 7- and 28-day survival, our study contributes complementary insights to the understanding of sepsis immunopathogenesis and outcome prediction.

## 2. Subjects and Methods

### 2.1. Subjects and Sample Collection

Our study comprised 43 patients (mean age 58.977 ± 12.229 years; 28 men and 15 women) hospitalized at the intensive care unit (ICU) of the 1st Department of Anesthesiology and Intensive Care Medicine, Faculty of Medicine, Comenius University and University Hospital Bratislava, Slovakia with a diagnosis of systemic inflammatory response (sepsis, septic shock or non-infectious SIRS) or for suspicion of this diagnosis (Table 1). Inclusion criteria were age of at least 18 years and expected length of hospitalization of more than 24 h. Exclusion criteria included primary immunodeficiency or the use of immunosuppressive treatment in the anamnesis. Terminally ill patients with an unfavorable prognosis predicting death within 24 h of admission to ICU were also excluded.

All patients underwent a standard entrance examination by the ICU physician and routine laboratory examinations including complete and differential blood count, levels of inflammatory markers, biochemical examinations, blood culture and other microbiological examinations according to indication. Baseline clinical data collected included: sex, age, body mass index (BMI), comorbidities, APACHE II (Acute Physiology and Chronic Health Evaluation II; a severity-of-disease classification system, one of several ICU scoring systems) and SOFA clinical score (sequential organ failure assessment score, previously known as the sepsis-related organ failure assessment score—determine the extent of a person’s organ function or rate of failure. The score is based on six different values, one for the respiratory, cardiovascular, hepatic, coagulation, renal and neurological systems within the first 24 h. We monitored also duration of mechanical ventilation, length of stay in ICU and the survival rate at 7th a 28th day. Patients were followed until discharge from the ICU or death. Regarding clinical status, group of septic patients acquired significantly higher APACHE II scores (*p* = 0.0033). They also scored more in the case of SOFA score, but without a statistically significant difference (*p* = 0.0906) (Table 1). Antimicrobial treatment was initiated in all patients according to current clinical standards and was further individualized based on the results of microbiological investigations.

10 mL of peripheral blood was collected from each patient on admission and 5 mL subsequently during the first 24 h and on the 3rd, 5th, 7th, and 10th day of hospitalization, if the patient survived, or his health did not improve enough to be discharged from the ICU. Blood was collected in tubes with EDTA (ethylenediaminetetraacetic acid) in order to isolate the plasma using centrifugation (2500 rpm, 15 min). The plasma samples were immediately divided into aliquots and subsequently frozen and stored at −80 °C in deep freezer box. For the purposes of this project, blood and plasma samples obtained during the first 24 h of hospitalization in the ICU were used. Although laboratory personnel were blinded to patients’ clinical data and subgroup allocation, complete blinding of the study was not feasible, which represents a methodological limitation.

All the investigations were carried out in accordance with the International Ethical Guidelines and the Declaration of Helsinki. Written informed consent for enrolling in the study and for personal data management was obtained from all examined cases. The study was approved by the Ethics Committee of the University Hospital in Bratislava, Ethics Committee approval No. 56/2014 (extension of the approval by the Ethics Committee through 2026 No. 247/2018). Blood samples were collected from patients between the years 2014 and 2018.

### 2.2. Laboratory Analyses

The routine hematological laboratory examinations that were carried out in the hospital laboratory Medirex, a.s. at the University Hospital Bratislava, Ruzinov, includes determination of complete and differential blood count. In the same laboratory, standard biochemical examinations were also carried out including the determination of plasma concentrations of presepsin (sCD14-ST) by the PATHFAST method—Presepsin chemiluminescent enzyme immunoassay (CLEIA; Mitsubishi Chemical Europe GmbH, Dusseldorf, Germany) using an automatic analyzer, PCT by the chemiluminescent method immunoassays (electrochemiluminiscent enzyme immunoassay; ECLEIA) and C-reactive protein (CRP) by immunoturbidimetry.

#### 2.2.1. Humoral Pro- and Anti-Inflammatory Biomarkers sTREM-1, HMGB1, IL-6, IL-10, sHLA-G

Plasma concentrations of soluble markers of inflammation and cytokines sTREM-1, HMGB1, IL-6, IL-10, sHLA-G were determined in the laboratories of the Institute of Immunology of the Faculty of Medicine, Comenius University in Bratislava, Slovakia using the sandwich ELISA (Enzyme-linked Immunosorbent Assay) method; human sTREM-1 ELISA kit (Cloud-Clone Corp., Houston, TX, USA), human HMGB1 ELISA kit (Wuhan Fine Biotech Co., Ltd., Wuhan, China), human IL-6 ELISA kit (Wuhan Fine Biotech Co., Ltd., Wuhan, China), human IL-10 ELISA kit (Wuhan Fine Biotech Co., Ltd., Wuhan, China), sHLA-G ELISA kit (BioVendor—Laboratorní medicína, Ltd. and EXBIO Praha, Ltd., Prague, Czech Republic). We proceeded strictly in accordance with manufacturer’s instructions. The concentrations of the reaction product were measured using a spectrophotometer at a wavelength of 450 nm.

#### 2.2.2. Plasma Concentration of 25-hydroxyvitamin D (25(OH)D)

25-hydroxyvitamin D is the major circulating form of VD, which has a half-life of approximately 2-3 weeks. It includes two forms of VD, VD received in the diet and VD produced by exposure of the skin to sunlight. The concentration of 25(OH)D in plasma was determined by the electrochemiluminescence binding test (Elecsys Vitamin D total-Cobas; Roche Diagnostics GmbH, Mannheim, Germany) using an immunochemical analyzer at Laboratoria Piestany, Ltd., Piestany, Slovakia. VD deficiency was defined as a plasmatic 25(OH)D level < 20 ng/mL, VD insufficiency 25(OH)D level between 20 and 29 ng/mL, preferred level for 25(OH)D > 30 ng/mL and a severe VD deficiency was defined as a plasmatic 25(OH)D level < 12 ng/mL [27].

## 3. Statistics

We subjected the obtained data to statistical analysis using the statistical program InStat3.06 (GraphPad Software, Inc., San Diego, CA USA) and SAS EG (8.4) (SAS Institute, Inc., 100 SAS Campus Drive, Cary, North Carolina, 27513, USA). We compared the values using the parametric *t*-test or the non-parametric Mann-Whitney test if the data did not pass the normality test. We used the non-parametric Spearman test for correlation. To control for type I error due to multiple comparisons, the False Discovery Rate (FDR) correction was applied using the Benjamini-Hochberg procedure, and *p*-values were adjusted accordingly. We considered *p* values < 0.05 to be statistically significant. Logistic regression analysis was used to account for potential confounding effects of age, sex, BMI, and comorbidities. Survival analysis was performed using Kaplan–Meier curves, log-rank and Wilcoxon tests, as well as Cox proportional hazards regression adjusted for clinical covariates.

## 4. Results

### 4.1. Comparison of Septic and Non-Infectious SIRS Patients

Significant differences were observed between septic patients (sepsis and septic shock) and those with non-infectious SIRS in several inflammatory and immune markers. Septic patients had markedly higher levels of sTREM-1, CRP, PCT, and sCD14-ST, with all differences remaining statistically significant after false discovery rate (FDR) correction (adjusted *p* < 0.05). These findings are consistent with the known pathophysiology of sepsis, which involves a more pronounced inflammatory response.

Additionally, vitamin D levels were significantly lower in septic patients compared to those with non-infectious SIRS (adjusted *p* = 0.0247), suggesting a potential role of vitamin D deficiency in sepsis severity or susceptibility. The neutrophil-to-lymphocyte ratio (Neu/Ly) was also significantly elevated in the septic group (adjusted *p* = 0.0114), supporting its utility as a marker of systemic inflammation.

Other markers such as HMGB1, IL-6, IL-10, and sHLA-G did not show statistically significant differences between the groups after correction, indicating that their discriminatory power in this context may be limited (Table 2).

### 4.2. Logistic Regression Analysis of Most Predictive Biomarkers Distinguishing Septic (Sepsis + Septic Shock) and Non-Infectious SIRS Patients

Logistic regression analysis identified C-reactive protein (CRP) and procalcitonin (PCT) as the most predictive biomarkers for distinguishing sepsis/septic shock from non-infectious SIRS patients, with score chi-square values of 15.97 and 13.68, respectively (Table 3). ROC analysis confirmed these findings, with area under the curve (AUC) values of [0.9583] for CRP, [0.8125] for PCT (Figure 1).

Serum vitamin D levels remained a statistically significant predictor of sepsis diagnosis after adjusting for age, sex, BMI, and comorbidities in the multivariable logistic regression model (*p* = 0.025). Higher vitamin D levels were independently associated with a lower likelihood of sepsis (OR = 0.741), suggesting a potential protective effect. Additionally, soluble CD14 (sCD14-ST) and PCT were also identified as significant predictors (*p* = 0.045 and *p* = 0.044, respectively), supporting their diagnostic relevance in distinguishing sepsis from non-infectious systemic inflammation. In contrast, other biomarkers such as C-reactive protein (CRP) and triggering receptor expressed on myeloid cells-1 (TREM-1) did not reach statistical significance after adjustment for clinical covariates (*p* = 0.147 and *p* = 0.176, respectively), (Table 4). Interestingly, while CRP initially emerged as the strongest predictor of sepsis based on univariate logistic regression and ROC analysis (AUC = 0.9583), it lost statistical significance after adjustment for age, sex, BMI, and comorbidities (*p* = 0.147). In contrast, serum vitamin D levels remained a significant independent predictor (*p* = 0.025), with higher levels associated with a reduced likelihood of sepsis (OR = 0.741).

### 4.3. Comparison of Inflammatory Markers Between Survivors and Non-Survivors on Day 7

Statistical analysis revealed in patients surviving the 7th day of hospitalization significantly higher plasma concentrations of VD (*p* = 0.0035; after FDR correction: *p* < 0.0315), and lover concentrations of IL-10 (*p* < 0.0057; after FDR correction: *p* < 0.0257), (Table 5).

### 4.4. Comparison of Inflammatory Markers Between Survivors and Non-Survivors on Day 28

When comparing 28-day survival outcomes, most biomarkers did not show statistically significant differences between survivors and non-survivors after correction for multiple testing using the false discovery rate (FDR). Notably, IL-10 levels were higher in non-survivors (median: 711.95 ng/L) compared to survivors (median: 278.27 ng/L), with a nominal *p*-value of 0.0176; however, this difference did not remain significant after FDR correction (adjusted *p* = 0.1760). Similarly, vitamin D levels tended to be lower in non-survivors (median 9.105 μg/L) than in survivors (median 13.248 μg/L), with borderline nominal significance (*p* = 0.0500), but again not statistically significant after FDR adjustment (adjusted *p* = 0.2500) (Table 6).

### 4.5. Logistic Regression Analysis of 7-Day and 28-Day Survival

Logistic regression was performed to identify biomarkers associated with short-term (7-day) and intermediate-term (28-day) survival. For 7-day survival, IL-10 emerged as the strongest individual predictor (Score Chi-Square = 12.56), followed by vitamin D (VD, Score Chi-Square = 5.67). The best two-variable model included HMGB1 and IL-10 (Score Chi-Square = 14.38. For 28-day survival, IL-10 again showed the highest individual predictive value (Score Chi-Square = 9.00), with VD as the next best marker (Score Chi-Square = 5.04). The best two-variable model combined IL-10 and VD (Score Chi-Square = 10.37, (Table 7).

### 4.6. Receiver Operating Characteristic Analysis for 7-Day and 28-Day Survival

Receiver operating characteristic (ROC) analysis was performed to evaluate the predictive value of IL-10 for survival at 7 and 28 days. IL-10 demonstrated good discriminative ability for 7-day survival (AUC = 0.8284) and moderate predictive performance for 28-day survival (AUC = 0.7793), (Figure 2).

Receiver operating characteristic (ROC) analysis was performed to assess the predictive value of vitamin D for 7-day and 28-day survival. Vitamin D showed good discriminative ability for 7-day survival (AUC = 0.8243), but only modest predictive performance for 28-day survival (AUC = 0.6818), (Figure 3).

### 4.7. Relationship Between Biomarker Levels and 7-Day and 28-Day Survival

To assess the relationship between biomarker levels and short-term (7-day) and intermediate-term (28-day) survival, logistic regression analyses were performed with adjustment for age, sex, BMI, and comorbidities. Following FDR correction, significant differences between survivors and non-survivors at 7 days were observed for IL-10 and serum vitamin D levels. No other biomarkers showed statistically significant differences at this time point. For 28-day survival, no significant differences remained after FDR correction.

To further evaluate the robustness of these findings, multivariable logistic regression was applied to IL-10 and vitamin D. After adjustment for clinical covariates, IL-10 remained a statistically significant predictor of both 7-day and 28-day survival (*p* = 0.037 and *p* = 0.029, respectively). Vitamin D showed a borderline positive association with 28-day survival (*p* = 0.071), suggesting a potential but non-definitive prognostic value. The direction of the regression coefficients (β) further supports the biological relevance of the findings. IL-10 showed a negative association with survival, indicating that higher levels were linked to increased mortality. In contrast, vitamin D exhibited a positive association, suggesting a potential protective effect (Table 8).

### 4.8. Kaplan–Meier Curves and Cox Proportional Hazards Regression Analysis for Prognostic Relevance of Biomarkers and Survival Analysis

Kaplan–Meier survival analysis stratified by IL-10 serum levels (<843.63 vs. ≥843.63 pg/mL) revealed a significantly higher survival probability in patients with IL-10 levels below the cut-off. The optimal threshold (843.63 pg/mL) was determined using the Youden index from ROC analysis and confirmed by log-rank and Wilcoxon tests (χ^2^ = 26.38 and 28.11, respectively; *p* < 0.0001) (Figure 4).

To account for potential confounding factors, a Cox proportional hazards model was applied, including age, sex, BMI, and comorbidities. Among the tested variables, IL-10 remained a statistically significant predictor of survival (Hazard Ratio [HR] = 1.002; *p* = 0.0096), indicating that higher IL-10 levels were positively associated with increased mortality risk. Vitamin D showed a borderline association with survival (HR = 0.868; *p* = 0.0635), suggesting a potential protective effect. None of the clinical covariates demonstrated significant associations with survival in the adjusted model (Table 9).

### 4.9. Correlations of sTREM-1 with Other Estimated Parameters

By investigating the relationship of the soluble form of TREM-1 with other markers of inflammation and variables characterizing the clinical condition of patients with a systemic inflammatory response of both infectious and non-infectious etiology, we revealed a positive correlation of sTREM-1 concentration with the level of CRP (*p* < 0.0001), PCT (*p* = 0.0021), sCD14-ST (*p* = 0.0096), also with the number of comorbidities present in patients (*p* = 0.0032) and the clinical scores of APACHE II (*p* = 0.0014) and SOFA (*p* = 0.0115). There was also an indicated, albeit statistically insignificant, positive correlation with plasma HMGB1 concentration (*p* = 0.0646). On the contrary, we found a negative correlation of sTREM-1 level with the plasma concentration of vitamin D (*p* = 0.0049) (Table 10).

### 4.10. Correlations of HMGB1 with Other Estimated Parameters

The concentration of HMGB1 positively correlated with concentrations of CRP (*p* = 0.0149) and PCT (*p* = 0.0196), not significantly with the concentration of sTREM-1 (*p* = 0.0646). A negative correlation was present in the case of the immunosuppressive and anti-inflammatory molecule sHLA-G (*p* = 0.0190) and the absolute number of leukocytes (*p* = 0.0194) and neutrophils (*p* = 0.0173), and borderline, statistically insignificant in the case of the absolute number of lymphocytes (*p* = 0.0776) and monocytes (*p* = 0.0642) (Table 10).

### 4.11. Correlations of Cytokines IL-6 and IL-10 with Other Estimated Parameters

The concentration of proinflammatory cytokine IL-6 positively correlated with concentration of plasma PCT (*p* = 0.0315) and statistically insignificantly with concentration of CRP (*p* = 0.0639). There was also a statistically significant negative correlation with VD (*p* = 0.0325) (Table 10). The plasma level of anti-inflammatory and immunoregulatory IL-10 significantly positively correlated with SOFA score (*p* = 0.0054), however, insignificantly with APACHE II (*p* = 0.0711) and patients’ age (*p* = 0.0819) (Table 10).

### 4.12. Correlations of the Level of Anti-Inflammatory and Immunosuppressive Molecules sHLA-G and 25(OH) Vitamin D

A negative correlation of the plasma concentration of the immunosuppressive molecule sHLA-G with the concentration of HMGB1 (*p* = 0.0190), the absolute number of eosinophils in the peripheral blood (*p* = 0.0150) and the clinical SOFA score (*p* = 0.0274) (Table 10) were observed.

We also obtained interesting results by examining the relationship between anti-inflammatory VD and other indicators. We revealed a negative correlation of VD plasma concentration with the concentrations of inflammatory biomarkers CRP (*p* = 0.0007), sCD14-ST (*p* = 0.0041), sTREM-1 (*p* = 0.0049), IL-6 (*p* = 0.0325), and clinical SOFA score (*p* = 0.0431). A negative correlation with PCT (*p* = 0.0807) and APACHE II (0.0697) was indicated but statistically insignificant. The concentration of VD in the plasma was further positively correlated with the absolute number of neutrophils (*p* = 0.0164) and monocytes (*p* = 0.0461) in the peripheral blood, statistically insignificantly with the total number of leukocytes (*p* = 0.0749) (Table 10).

## 5. Discussion

According last Sepsis 3 definition, sepsis is defined as life-threatening organ dysfunction caused by a dysregulated host response to infection [1]. However, pathogenetic processes associated with SIRS and CARS remains accepted. Early diagnosis of sepsis and immediate initiation of antimicrobial therapy is essential to reduce mortality of sepsis. Since the life of a patient with sepsis is threatened by both excessive inflammation and immunosuppression even immune paralysis, in addition to the level of pro- and anti-inflammatory cytokines, it is necessary to monitor the state of immunity (Th1/Th2) [8,9,28]. The aim of our study was to try to diagnose sepsis early by examining a combination of molecules that are involved in the pathogenesis of sepsis. We examined the inflammatory markers sTREM-1, HMGB1, sCD14-ST and the Neu/Ly ratio, but also the level of the anti-inflammatory and immunoregulatory Th2 cytokine IL-10 and the immunosuppressive and anti-inflammatory molecule HLA-G. The mentioned molecules are involved in the pathogenesis of sepsis.

When comparing septic patients with those with non-infectious SIRS, we found significantly elevated levels of several inflammatory markers in the sepsis group after FDR correction: sTREM-1 (adjusted *p* = 0.0138), CRP (adjusted *p* < 0.0010), PCT (adjusted *p* = 0.0023), sCD14-ST (adjusted *p* = 0.0005), and VD deficiency (adjusted *p* = 0.0247). Additionally, the Neu/Ly ratio was significantly higher in septic patients (adjusted *p* = 0.0114), supporting its utility as a marker of systemic inflammation and critical state of patients.

Correlation analysis revealed that plasma sTREM-1 levels were positively associated with CRP (adjusted *p* = 0.004), PCT (adjusted *p* = 0.021), sCD14-ST (adjusted *p* = 0.0384), number of comorbidities (adjusted *p* = 0.0256), and clinical severity scores APACHE II (adjusted *p* = 0.0187) and SOFA (adjusted *p* = 0.0418). A significant negative correlation between sTREM-1 and vitamin D levels (adjusted *p* = 0.0245) suggests a potential immunomodulatory role of vitamin D in sepsis.

These findings underscore the diagnostic and prognostic relevance of sTREM-1 in sepsis. Its strong association with both inflammatory markers and clinical severity supports its use not only in early diagnosis but also in monitoring disease progression. The positive correlation with comorbidities may reflect heightened inflammatory reactivity or impaired immune regulation in patients with chronic conditions, consistent with previous studies [29,30]. Other published works also identified sTREM-1 as a promising marker for the early diagnosis of sepsis and highlight its potential in combination with other biomarkers to become a reference diagnostic test in sepsis [31,32].

The Neu/Ly ratio—marker of systemic inflammation and critical state and riskiness of patients, was significantly higher (adjusted *p* = 0.0114) in the peripheral blood of patients with sepsis. Rapidly developed lymphopenia observed in patients with sepsis indicates significant immunosuppression and is also present in severe “non-infectious” conditions such as acute trauma, extensive burns or major operations [33]. Lymphopenia in sepsis subsequently affects the individual’s immune response on several levels and is associated with a higher risk of infection and mortality in various clinical settings [34,35]. The question remains whether, in the case of the investigated group of septic patients, it is a consequence of ongoing sepsis, or lymphopenia was one of the causes of the development of a systemic infection. We excluded the influence of corticoid administration on the absolute number of lymphocytes in patients with sepsis by comparing the number of lymphocytes in the subgroup of patients who received corticoids with the subgroup without the mentioned treatment (*p* = 0.1187).

Despite the high prevalence of hypovitaminosis D in critically ill patients, VD is often overlooked in acute care settings. In our cohort, septic patients had at the time in ICU admission significantly lower VD levels compared to those with non-infectious SIRS (adjusted *p* = 0.0247). Importantly, the level of VD remained a statistically significant independent predictor of sepsis in multivariable logistic regression (*p* = 0.025), with higher levels associated with a reduced risk (OR = 0.741). This contrasts with CRP, which despite excellent discriminatory power in univariate analysis (AUC = 0.9583), lost its significance after adjustment for clinical covariates (*p* = 0.147), likely due to its nonspecific elevation in chronic inflammatory states.

These results support the hypothesis that VD may exert its protective anti-inflammatory and immunomodulatory effect, influencing both susceptibility to infection and the severity of systemic inflammation. The plasma concentration of VD showed significant negative correlations with inflammatory markers such as CRP (adjusted *p* = 0.014), sCD14-ST (adjusted *p* = 0.0273), sTREM-1 (adjusted *p* = 0.0245), and IL-6 (adjusted *p* = 0.0542), as well as with the SOFA score (adjusted *p* = 0.0639). Additionally, VD levels positively correlated with the absolute number of neutrophils (adjusted *p* = 0.0469) and monocytes (adjusted *p* = 0.0659), suggesting a link between VD status and innate immune cell availability. Taken together, these findings highlight the importance of integrating both inflammatory and immunoregulatory biomarkers in sepsis diagnostics. VD deserves further investigation both as a prognostic marker and potential therapeutic target in systemic inflammatory conditions. Nevertheless, causality cannot be inferred from this observational study, and randomized supplementation trials will be required to determine whether VD exerts a therapeutic effect.

We obtained interesting results during monitoring the significance of the examined biomarkers with patient survival. Our findings highlight the important role of particularly IL-10 and VD, in predicting short- (7 day) and intermediate-term (28 day) mortality in hospitalized patients. When comparing patients who survived seven days of hospitalization with those who did not, significantly lower IL-10 levels and higher VD concentrations were observed. In the group of survivors, higher concentrations of sHLA-G and sCD14-ST were also noted, although these differences did not reach statistical significance. Conversely, patients who did not survive consistently exhibited higher IL-10 values. A similar pattern was observed at the 28-day follow-up, where survivors had higher VD concentrations, lower IL-10 values, and slightly elevated sHLA-G concentrations.

IL-10 is a potent immunoregulatory and anti-inflammatory cytokine that suppresses Th1 immunity (Tc cells, NK cells, macrophages) and inhibits the production of pro-inflammatory cytokines, such as IFN-γ, TNF-α, IL-1β, IL-6, IL-8, and IL-12. It is secreted by various cell types, including Th2 lymphocytes, M2 macrophages, dendritic cells, and epithelial cells [36]. In our study, higher IL-10 levels were consistently associated with increased mortality both in the early period (up to 7 days) and at the intermediate-term follow-up (up to 28 days). Elevated IL-10 values within the first 24 h after ICU admission may reflect an early onset of immunosuppression. This aligns with prior work confirming IL-10 as a predictor of short- and long-term mortality in sepsis [37,38,39], although other studies reported conflicting findings possibly due to methodological differences [40,41]. Importantly, IL-10 levels in our study positively correlated with the SOFA score, underscoring its association with the severity of organ dysfunction. In contrast, higher VD levels in survivors suggest a potential protective effect. Our findings are consistent with previous studies demonstrating that VD deficiency at ICU admission is an independent predictor of overall mortality in critically ill patients, regardless of comorbidities [42,43]. This relationship was further supported in our cohort by a significant correlation between VD and SOFA score, but not with APACHE II, which incorporates chronic comorbidities alongside acute illness severity. These results support the notion that VD status reflects the acute inflammatory and organ dysfunction burden more than chronic disease background. Experimental evidence also reinforces the protective potential of VD.

In an animal model of sepsis, Rao et al. (2018) demonstrated that VD treatment improved survival in LPS-induced sepsis by inhibiting HMGB1 secretion, a key late mediator of sepsis. Mechanistically, VD attenuated LPS-induced HMGB1 release by blocking its nuclear-to-cytoplasmic translocation in macrophages [44]. These findings resonate with our observation of more pronounced VD deficiency alongside slightly elevated HMGB1 levels in septic patients, suggesting a complex interplay between VD status, HMGB1 secretion, and systemic inflammation.

Kaplan–Meier and Cox regression analyses further confirmed IL-10 as an independent predictor of mortality, while VD demonstrated only a borderline protective effect. In addition, the trends observed with sHLA-G, sCD14-ST, and HMGB1 highlight potential additional biomarkers worthy of further exploration. Larger, prospective studies are warranted to validate these findings and to assess whether therapeutic modulation of VD status or IL-10 regulation could improve outcomes in critically ill patients.

HLA-G is an immune checkpoint molecule with immunosuppressive and anti-inflammatory activities [45]. As a tolerogenic molecule (either membrane bound or soluble (sHLA-G) broadly regulates both innate and adaptive immunity, as well as inflammatory processes [46]. HLA-G can inhibit a wide range of immunocompetent cells by interacting with T and B lymphocytes, NK cells, and polymorphonuclear leukocytes at all stages of the immune response—cell differentiation, proliferation, cytolysis, cytokine secretion, and immunoglobulin production [47].

In our study, plasma concentrations of sHLA-G were higher in patients who survived both the 7th and the 28th day of hospitalization, although the differences did not reach statistical significance after FDR correction. Nevertheless, correlation analyses revealed significant negative associations between sHLA-G plasma levels and pro-inflammatory HMGB1 (*p* = 0.0475), eosinophil counts (*p* = 0.0462), and SOFA scores (*p* = 0.0548, borderline significance). These results suggest that higher sHLA-G levels are linked with a more favorable clinical status and better survival outcomes. Considering its known immunomodulatory properties, sHLA-G may play a protective immunoregulatory role in patients with systemic inflammatory response, which has also been described by other authors [48]. However, we did not observe a significant difference in sHLA-G levels between patients with sepsis and those with non-infectious SIRS.

HMGB1 is a non-histone intranuclear protein that, during inflammation, is actively secreted by activated monocyte–macrophage system cells, or passively released from necrotic and damaged cells, acting as an “alarmin.” It promotes dendritic cell migration and maturation and supports polarization of naïve T-lymphocytes into the Th1 subpopulation. As a late pro-inflammatory cytokine, HMGB1 appears in circulation 16–32 h after endotoxin stimulation of immune cells [49]. In our study, plasma concentrations of HMGB1 did not differ significantly between survivors and non-survivors at either day 7 or day 28. However, correlation analyses revealed significant positive associations with CRP (*p* = 0.0497) and PCT (*p* = 0.0413), and significant negative correlations with sHLA-G (*p* = 0.0475), total leukocyte count (*p* = 0.0431), and neutrophil count (*p* = 0.0461). These findings suggest that HMGB1 is closely linked to the intensity of systemic inflammation and immune dysregulation. Grégoire et al. (2017) demonstrated in a murine model and in septic shock survivors that HMGB1 contributes to neutrophil dysfunction, specifically impairing NADPH oxidase activity and neutrophil-dependent bacterial clearance [50]. Anti-HMGB1 treatment preserved neutrophil function and improved bacterial clearance. Based on these observations, HMGB1 accumulation in the later stages of sepsis may represent a marker of post-septic immunosuppression and could predict late mortality due to immune paralysis. However, in our study, plasma HMGB1 concentrations measured at ICU admission had no prognostic value.

The main limitation of our study is the relatively small and heterogeneous cohort (*n* = 43), which reduces statistical power and limits the generalizability of our findings. In addition, the study was conducted at a single center, which may introduce center-specific biases. Finally, long-term follow-up data beyond the mid-term survival analysis were not available, which limits conclusions regarding the sustained prognostic value of the biomarkers. These factors should be taken into account, and our results should be validated in larger, multicenter cohorts with extended follow-up.

## 6. Conclusions

Our study confirms CRP, PCT, and sCD14-ST as the most reliable biomarkers for distinguishing sepsis from non-infectious SIRS, with sTREM-1 providing additional, though non-independent, diagnostic and monitoring value. Importantly, multivariable regression analysis revealed that CRP, despite its excellent univariate discriminatory power, lost significance after adjustment for demographic and clinical covariates. In contrast, VD remained an independent protective factor, underscoring its potential pathophysiological and therapeutic relevance. Elevated IL-10 levels were consistently associated with increased 7-day and 28-day mortality, while higher VD concentrations correlated with better survival, highlighting their complementary prognostic roles. Although sHLA-G and HMGB1 did not independently predict outcomes, their associations with immune dysregulation and inflammation suggest further research potential. Together, these findings emphasize that combining classical inflammatory markers with immunoregulatory biomarkers may enhance diagnostic accuracy and risk stratification in sepsis, a strategy that requires validation in larger multicenter cohorts.

## Figures and Tables

**Figure 1 biomedicines-13-02481-f001:**
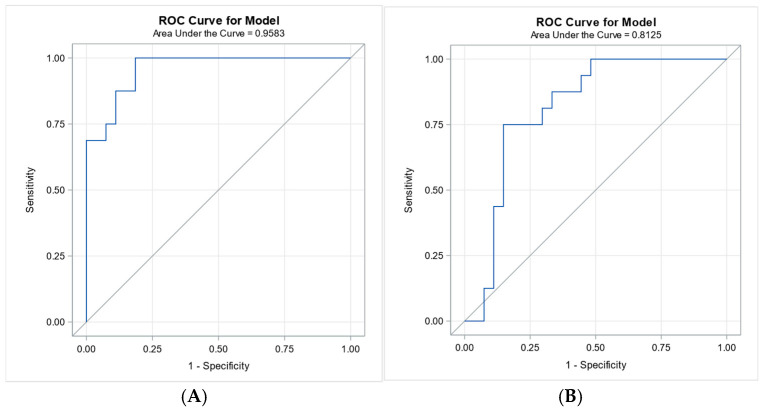
Receiver operating characteristic (ROC) curves for CRP and PCT in distinguishing sepsis/septic shock from SIRS. Legend: Panel (**A**): ROC curve for C-reactive protein (CRP) with an area under the curve (AUC) of 0.9583, indicating excellent discriminative ability. Panel (**B**): ROC curve for procalcitonin (PCT) with an AUC of 0.8125, demonstrating good discriminative performance. Both biomarkers were identified as top predictors by logistic regression analysis.

**Figure 2 biomedicines-13-02481-f002:**
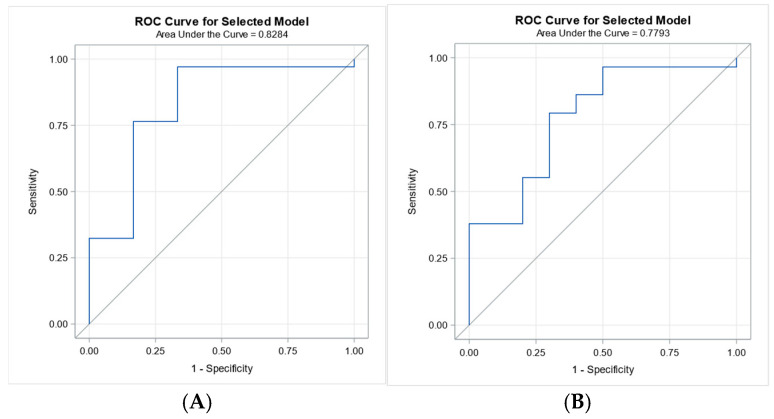
ROC curves for IL-10 in predicting 7-day and 28-day survival. Legend: Panel (**A**): ROC curve for IL-10 predicting 7-day survival (AUC = 0.8284), demonstrating good predictive accuracy. Panel (**B**): ROC curve for IL-10 predicting 28-day survival (AUC = 0.7793), indicating moderate predictive performance.

**Figure 3 biomedicines-13-02481-f003:**
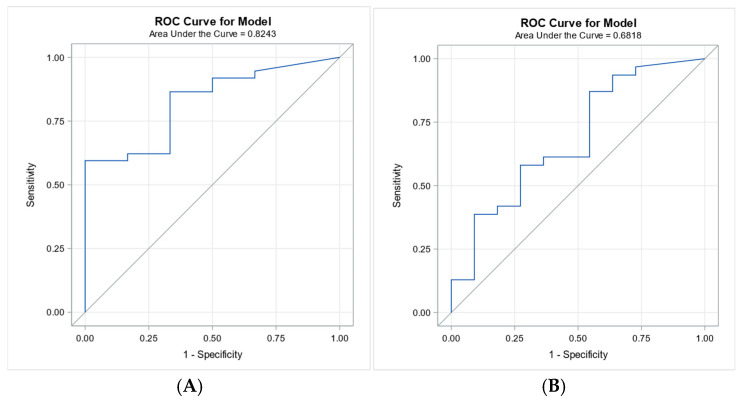
ROC curves for vitamin D in predicting 7-day and 28-day survival. Legend: Panel (**A**): ROC curve for vitamin D predicting 7-day survival (AUC = 0.8243), demonstrating good predictive accuracy. Panel (**B**): ROC curve for vitamin D predicting 28-day survival (AUC = 0.6818), indicating modest predictive performance.

**Figure 4 biomedicines-13-02481-f004:**
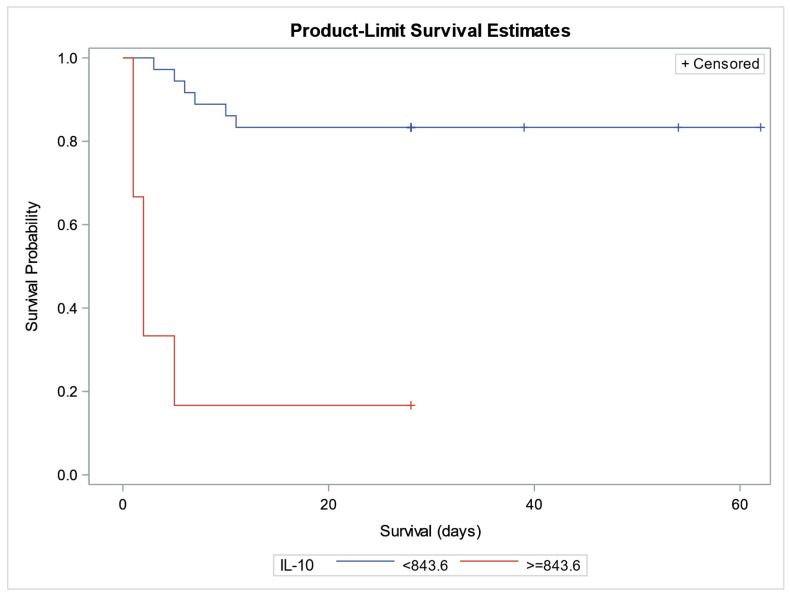
Kaplan–Meier survival curves stratified by IL-10 serum levels (pg/mL). Legend: Kaplan–Meier survival analysis comparing patients with IL-10 levels < 843.6 pg/mL (blue curve) and ≥843.6 pg/mL (red curve). The x-axis represents survival time in days; the y-axis shows survival probability. “+” symbols indicate censored observations. Patients with higher IL-10 levels (≥843.6 pg/mL) demonstrated significantly improved survival.

**Table 1 biomedicines-13-02481-t001:** Characteristics of the investigated individuals.

	All Patients	SIRS	Sepsis	Septic Shock
Number	43	16	11	16
Mean age (years)	58.977 ± 12.229	56.875 ± 16.342	60.545 ± 9.554	60.0 ± 9.121
Sex (male/female)	28/15	10/6	8/3	10/6
Comorbidities:				
NYHA III-IV	15	7	5	3
AH	27	9	8	10
COPD	2	0	1	1
Diabetes mellitus	12	1	4	7
CHRI	5	2	2	1
Hepatopathy	7	1	3	3
Malignancy	4	0	2	2
APACHE II	30.395 ± 9.396	24.563 ± 9.899	32.000 ± 8.173	35.125 ± 6.531
SOFA	12.674 ± 2.146	11.938 ± 2.175	12.273 ± 1.902	13.688 ± 1.991

Legend: SIRS—systemic inflammatory response syndrome; NYHA—New York Heart Association of heart failure; COPD—chronic obstructive pulmonary disease; CHRI—chronic renal insufficiency; AH—arterial hypertension; APACHE II-acute physiology and chronic health evaluation II; SOFA—sequential organ failure assessment score.

**Table 2 biomedicines-13-02481-t002:** Comparison of biomarker concentrations in a group of septic patients (sepsis and septic shock) with a group of patients with non-infectious SIRS.

Marker	Group	N	Median/Mean	IQR/SD	*p* (Mann-Whitney/*t*-Test with Welch’s Correction *)	Adjusted *p*-Value (FDR Correction)
sTREM-1 (ng/L)	SIRS	16	96.512	75.083	0.0055	0.0138
	sepsis	27	201.48	154.71		
HMGB1 (ng/L)	SIRS	14	788.12	243.18	0.0951 *	0.1359
	sepsis	26	919.55	194.33		
CRP (mg/L)	SIRS	16	62.120	35.654	<0.0001 *	<0.0010
	sepsis	27	265.17	155.63		
PCT (ng/L)	SIRS	16	10.094	14.291	0.0007	0.0023
	sepsis	27	53.593	40.992		
sCD14-ST (ng/L)	SIRS	16	579.75	637.37	0.0001	0.0005
	sepsis	27	2399.0	5232.5		
IL-6 (ng/L)	SIRS	14	187.80	149.45	0.7430	0.7430
	sepsis	20	242.64	220.93		
IL-10 (ng/L)	SIRS	16	296.29	262.45	0.7156	0.7951
	sepsis	27	443.95	500.11		
VD (μg/L)	SIRS	16	15.719	7.382	0.0148 *	0.0247
	sepsis	27	10.246	4.856		
sHLA-G (U/mL)	SIRS	16	48.159	55.134	0.4435	0.5544
	sepsis	27	58.872	50.553		
Neu/Ly	SIRS	16	8.125	5.303	0.0057	0.0114
	sepsis	25	17.243	14.218		

Legend: sTREM-1-soluble triggering receptor expressed on myelocytes; HMGB-1-high mobility group box 1 protein; CRP-C-reactive protein; PCT-procalcitonin; sCD14-ST-presepsin; IL-6-Interleukin-6; IL-10-interleukin-10; VD-25-hydroxyvitamin D (25(OH)D); sHLA-G-soluble human leukocyte antigen-G; Ly-lymphocytes; Neu-neutrophils, N-number; IQR-interquartile range; SD-standard deviation; FDR-false discovery rate; *p*-value after FDR correction < 0.05 was considered significant.

**Table 3 biomedicines-13-02481-t003:** Top-performing models by score Chi-Square (best subsets selection).

Number of Variables	Variables Included in Model	Score Chi-Square
1	CRP	15.97
1	PCT	13.68
2	CRP, PCT	19.99
3	CRP, PCT, Ne/Ly	20.57
4	CRP, PCT, sCD14-ST, VD (μg/L)	20.19
5	sTREM1, CRP, PCT, sCD14-ST, Ne/Ly	20.89

Legend: Best Subsets Selection identified several high-performing models for distinguishing sepsis/septic shock from SIRS. The model with two variables—CRP and PCT—was selected as the optimal balance between predictive performance and model simplicity. CRP and PCT were consistently present in all top models, highlighting their importance as key biomarkers.

**Table 4 biomedicines-13-02481-t004:** Logistic regression—biomarker predictors of sepsis.

Biomarker	Estimate (β)	Standard Error	*p*-Value	Interpretation
Vitamin D	–0.3004	0.1343	0.025	Significant, protective effect
sCD14	0.00188	0.00094	0.045	Significant positive predictor
PCT	0.0414	0.0205	0.044	Significant positive predictor
CRP	0.2579	0.1778	0.147	Not significant
sTREM-1	0.00799	0.0059	0.176	Not significant

Note: Multivariable logistic regression adjusted for age, sex, BMI, and comorbidities. Legend: β-regression coefficients; sCD14-presepsin; PCT-procalcitonin; CRP-C-reactive protein; sTREM-soluble triggering receptor expressed on myelocytes; *p*-value < 0.05 is considered significant.

**Table 5 biomedicines-13-02481-t005:** Comparison of the group of patients with systemic inflammatory reaction (septic + non-infectious SIRS together) who survived the 7th day of hospitalization with the group of patients who did not survive the 7th day of hospitalization.

Marker	7th Day Survival	N	Median/Mean	IQR/SD	*p* (Mann-Whitney/*t*-Test with Welch’s Correction *)	Adjusted *p*-Value (FDR Correction)
sTREM-1 (ng/L)	+	37	147.42	111.45	0.3096	0.3483
	−	6	254.92	249.28		
HMGB1 (ng/L)	+	34	858.77	227.99	0.1951 *	0.2508
	−	6	957.30	145.02		
CRP (mg/L)	+	37	178.90	163.17	0.1298	0.1947
	−	6	255.74	121.68		
PCT (ng/L)	+	37	37.195	40.364	0.4833	0.4833
	−	6	38.712	37.674		
sCD14-ST (ng/L)	+	37	1734.2	4559.3	0.0704	0.1267
	−	6	1647.7	841.92		
IL-10 (ng/L)	+	37	291.98	315.36	0.0057	0.0257
	−	6	987.36	579.38		
VD (μg/L)	+	37	13.205	6.331	0.0035 *	0.0315
	−	6	6.597	3.550		
sHLA-G (U/mL)	+	37	59.471	54.438	0.0546	0.1638
	−	6	26.613	13.677		
Eo (10^9^/L)	+	36	0.04944	0.09568	0.0583	0.1312
	−	4	0.2200	0.2855		

Legend: sTREM-1-soluble triggering receptor expressed on myelocytes; HMGB-1-high mobility group box 1 protein; CRP-C-reactive protein; PCT-procalcitonin; sCD14-ST-presepsin; IL-10-interleukin-10; VD-25-hydroxyvitamin D (25(OH)D); sHLA-G-soluble human leukocyte antigen-G; Eo-eosinophils; N-number; IQR-interquartile range; FDR-false discovery rate; SD-standard deviation; *p*-values after FDR correction < 0.05 were considered significant.

**Table 6 biomedicines-13-02481-t006:** Comparison of the group of patients with systemic inflammatory reaction (septic + non-infectious SIRS together) who survived the 28th day of hospitalization with the group of patients who did not survive the 28th day of hospitalization.

Marker	28th Day Survival	N	Median/Mean	IQR/SD	*p* (Mann-Whitney/ *t*-Test with Welch’s Correction *)	Adjusted *p*-Value (FDR Correction)
sTREM-1 (ng/L)	+	31	156.91	118.36	0.7530	0.8367
	−	11	186.10	195.48		
HMGB1 (ng/L)	+	29	859.93	228.02	0.5973 *	0.8533
	−	10	902.02	207.80		
CRP (mg/L)	+	31	197.21	167.17	0.9544	0.9544
	−	11	184.37	138.21		
PCT (ng/L)	+	31	43.256	41.284	0.3525	0.8813
	−	11	24.189	32.183		
sCD14-ST (ng/L)	+	31	1907.8	4956.9	0.4570	0.9140
	−	11	1327.8	999.92		
IL-6 (ng/L)	+	27	208.96	177.81	0.6326	0.7908
	−	6	305.89	255.50		
IL-10 (ng/L)	+	31	278.27	319.51	0.0176	0.1760
	−	11	711.95	561.34		
VD (μg/L)	+	31	13.248	6.534	0.0500 *	0.2500
	−	11	9.105	5.342		
sHLA-G (U/mL)	+	24	61.024	42.160	0.0614	0.2047
	−	11	47.573	56.588		
Eo (10^9^/L)	+	30	0.05667	0.1032	0.5591	0.9318
	−	9	0.1067	0.2059		

Legend: sTREM-1-soluble triggering receptor expressed on myelocytes; HMGB-1-high mobility group box 1 protein; CRP-C-reactive protein; PCT-procalcitonin; sCD14-ST-presepsin; IL-6-interleukin 6; IL-10-interleukin-10; VD-25-hydroxyvitamin D (25(OH)D); sHLA-G-soluble human leukocyte antigen-G; LEU-leukocytes; Eo-eosinophils; N-number; IQR-interquartile range; SD-standard deviation; FDR-false discovery rate; *p*-values after FDR correction < 0.05 were considered significant.

**Table 7 biomedicines-13-02481-t007:** Top-performing models by score Chi-Square (best subsets selection).

Survival	N	Markers	Score Chi-Square
7-day	1	IL-10	12.56
7-day	1	VD	5.67
7-day	2	HMGB1, IL-10	14.38
7-day	3	HMGB1, IL-10, VD (μg/L)	15.02
7-day	4	HMGB1, IL-10, sHLA-G, VD (μg/L)	15.27
28-day	1	IL-10	9
28-day	1	VD	5.04
28-day	2	IL-10, VD (μg/L)	10.37
28-day	3	PCT, IL-10, VD (μg/L)	12.21
28-day	4	HMGB1, PCT, IL-10, VD (μg/L)	13.29

Legend: Best subsets selection identified IL-10 as the most informative biomarker for both 7-day and 28-day survival. N-number of biomarkers.

**Table 8 biomedicines-13-02481-t008:** Multivariable logistic regression results for 7-day and 28-day survival.

Biomarker	Survival Time	Estimate (β)	Standard Error	*p*-Value	Odds Ratio (OR)	95% CI for OR	Interpretation
IL-10	7-day	–0.00492	0.00235	0.037	0.995	0.991–1.000	Significant predictor
IL-10	28-day	–0.00262	0.0012	0.029	0.997	0.995–1.000	Significant predictor
Vitamin D	28-day	0.1645	0.0912	0.071	1.179	0.986–1.409	Borderline significance

Note: Models adjusted for age, sex, BMI, and comorbidities. Legend: β-regression coefficients; CI-confidential interval; IL-10-interleukin.

**Table 9 biomedicines-13-02481-t009:** Cox proportional hazards regression analysis.

Variable	Estimate (β)	Standard Error	Hazard Ratio (HR)	*p*-Value	Interpretation
IL-10	0.00241	0.00093	1.002	0.0096	Significant predictor of increased mortality risk
Vitamin D	–0.14138	0.07619	0.868	0.0635	Borderline protective effect for survival
Age	–0.01617	0.04853	0.984	0.7389	Not significant
Sex (male vs. female)	–0.32945	1.11768	0.719	0.7682	Not significant
BMI	–0.03247	0.11118	0.968	0.7702	Not significant
NYHA III-IV	0.27215	0.96465	1.313	0.7778	Not significant
AH	1.3456	1.10075	3.841	0.2215	Not significant
COPD	–16.1199	4330	~0	0.997	Unstable estimate due to low event count
Diabetes	–0.31623	0.97971	0.729	0.7469	Not significant
CHRI	0.45045	1.14936	1.569	0.6951	Not significant
Hepatic Cirrhosis	1.14233	0.90567	3.134	0.2072	Not significant
Malignancy	–16.18316	2825	~0	0.9954	Unstable estimate due to low event count

Legend: β-regression coefficients; BMI-body mass index; NYHA-New York Heart Association of heart failure; COPD-chronic obstructive pulmonary disease; CHRI-chronic renal insufficiency; *p*-value < 0.05 is significant; AH-arterial hypertension.

**Table 10 biomedicines-13-02481-t010:** Correlations of markers of inflammation and variables characterizing the clinical condition of patients with a systemic inflammatory response of infectious and non-infectious etiology.

Marker		N	SR	95% CI	*p* (Spearman Test)	Adjusted *p*-Value (FDR Correction)
sTREM-1	CRP	43	0.5747	0.3233–0.7503	<0.0001	0.004
	PCT	43	0.4564	0.1719–0.6706	0.0021	0.021
	sCD14-ST	43	0.3908	0.09328–0.6242	0.0096	0.0384
	HMGB1	40	0.2950	−0.02774–0.5621	0.0646	0.078303
	VD	43	−0.4211	−0.6458–0.1291	0.0049	0.028
	Comorbid	43	0.4388	0.1504–0.6583	0.0032	0.0256
	APACHE II	43	0.4722	0.1913–0.6815	0.0014	0.018667
	SOFA	43	0.3820	0.08308–0.6179	0.0115	0.041818
HMGB1	CRP	40	0.3822	0.07066–0.6258	0.0149	0.049667
	PCT	40	0.3676	0.05374–0.6153	0.0196	0.041263
	sTREM-1	40	0.2950	−0.02774–0.5621	0.0646	0.076
	sHLA-G	40	−0.3694	−0.6166–0.05587	0.0190	0.0475
	Leu	40	−0.3683	−0.6158–0.05457	0.0194	0.043111
	Ly	38	−0.2898	−0.5646–0.04282	0.0776	0.081684
	Neu	38	−0.3840	−0.6327–−0.06344	0.0173	0.046133
	Mo	38	−0.3032	−0.5745–0.02811	0.0642	0.08025
IL-6	CRP	34	0.3213	−0.02940–0.6016	0.0639	0.088138
	PCT	34	0.3695	0.02538–0.6354	0.0315	0.057273
	VD	34	−0.3675	−0.6340–−0.02302	0.0325	0.056522
IL-10	Age	43	0.2683	−0.04407–0.5329	0.0819	0.0819
	APACHE II	43	0.2780	−0.03366–0.5403	0.0711	0.079
	SOFA	43	0.4172	0.1245–0.6431	0.0054	0.024
sHLA-G	HMGB1	40	−0.3694	−0.6166–0.05587	0.0190	0.044706
	Eo	40	−0.3819	−0.6256–0.07033	0.0150	0.046154
	SOFA	43	−0.3364	−0.5845–0.03091	0.0274	0.0548
VD	CRP	43	−0.4963	−0.6981–0.2215	0.0007	0.014
	PCT	43	−0.2694	−0.5337–0.04293	0.0807	0.082769
	sCD14-ST	43	−0.4294	−0.6517–0.1391	0.0041	0.027333
	sTREM-1	43	−0.4211	−0.6458–0.1291	0.0049	0.0245
	IL-6	34	−0.3675	−0.6340–−0.02302	0.0325	0.054167
	Leu	43	0.2744	−0.03746–0.5376	0.0749	0.080973
	Neu	41	0.3726	0.06393–0.6162	0.0164	0.046857
	Mo	41	0.3133	−0.003224–0.5728	0.0461	0.065857
	APACHE II	43	−0.2793	−0.5414–0.03217	0.0697	0.079657
	SOFA	43	−0.3100	−0.5647–0.001403	0.0431	0.063852

Legend: sTREM-1-soluble triggering receptor expressed on myelocytes; HMGB-1-high mobility group box 1 protein; IL-6-interleukin 6; IL-10-interleukin-10; sHLA-G-soluble human leukocyte antigen-G; VD-25-hydroxyvitamin D (25(OH)D); N-number; SR-Spearman r; CI-confidence interval; standard deviation; FDR-false discovery rate; *p*-values after FDR correction < 0.05 were considered significant.

## Data Availability

The raw data supporting the conclusions of this article are available from the authors upon reasonable request.

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
