# Peer review of "sTREM-1, HMGB1, CRP, PCT, sCD14-ST, IL-6, IL-10, sHLA-G, and Vitamin D in Relation to Clinical Scores and Survival in SIRS/Sepsis"

_biomedicines, 2025, doi:10.3390/biomedicines13102481_

Round 1
Reviewer 1 Report
Comments and Suggestions for Authors
The manuscript investigates systemic inflammatory biomarkers (sTREM-1, HMGB1, CRP, PCT, sCD14-ST, IL-6, IL-10, vitamin D, HLA-G, CD64 expression, etc.) in SIRS/sepsis patients and evaluates their correlation with APACHE II and SOFA scores as well as survival outcomes. Overall, the study addresses an important clinical challenge and provides relevant findings, but certain methodological and interpretative issues require clarification.
- Title is very long. Make it short and informative.
- The topic is timely and clinically relevant. The role of inflammatory biomarkers in improving early diagnosis and prognosis of sepsis remains an important unmet need, and the manuscript contributes valuable insights, particularly regarding sTREM-1, IL-10, and vitamin D.
- The abstract is comprehensive, but overly long and dense. It would benefit from greater conciseness with a clearer distinction between diagnostic and prognostic findings.
- The introduction provides good background on sepsis immunopathogenesis (SIRS, CARS, PICS), but some sections are lengthy and repetitive. Condensing this part would enhance readability without losing scientific depth.
- The rationale for including such a broad panel of biomarkers is well explained; however, the novelty compared with prior studies should be emphasized more clearly. Some cited literature overlaps with the current findings, so highlighting the distinct contribution of this cohort would strengthen the justification.
- The study population is relatively small (n=43) and heterogeneous. This limits statistical power and generalizability. The authors should acknowledge this limitation more explicitly and avoid over-interpretation.
- Inclusion and exclusion criteria are appropriate, but further detail on patient comorbidities and antimicrobial therapy would be useful since these factors may influence biomarker levels.
- The methods are generally sound, with clear descriptions of assays (ELISA, flow cytometry, routine labs). However, variability between assay kits from different manufacturers may affect comparability. Some comment on inter-assay reproducibility would be helpful.
- The choice of statistical tests is appropriate, but the results section contains a large number of comparisons. The risk of type I error should be discussed, and adjustments for multiple testing (e.g., Bonferroni or FDR correction) should be considered.
- The finding that IL-10 correlates with worse outcomes is interesting but requires caution—IL-10 kinetics may vary during sepsis progression. The authors should discuss the potential for serial measurements and dynamic changes rather than single time-point values.
- The role of HMGB1 is addressed, but its significance was limited in this cohort. The conclusion that HMGB1 might serve as a late-phase prognostic marker is reasonable but speculative, and this should be toned down.
- The results regarding TREM-1/TREM-2 cell expression are inconclusive due to low sample size. While the authors acknowledge this, it should be stressed that further validation is necessary.
- The emphasis on vitamin D deficiency as a risk factor and prognostic indicator is compelling and supported by correlations. However, causality cannot be established, and supplementation trials would be needed to confirm therapeutic implications.
- Figures and tables are informative but numerous. Consolidation of some data presentations (e.g., combining Tables 2–4 or summarizing correlations graphically) would improve clarity.
- The English language is overall understandable but requires editing for grammar, style, and conciseness. Certain sentences are long and could be simplified to improve readability.
- The conclusion is consistent with the results, but stronger caution is needed regarding overstatement of clinical applicability. The biomarkers identified (particularly sTREM-1, IL-10, vitamin D) are promising but not yet validated for routine practice.
- The references are adequate and up-to-date, but some could be shortened by focusing on the most relevant literature.
Author Response
Reviewer 1: The manuscript investigates systemic inflammatory biomarkers (sTREM-1, HMGB1, CRP, PCT, sCD14-ST, IL-6, IL-10, vitamin D, HLA-G, CD64 expression, etc.) in SIRS/sepsis patients and evaluates their correlation with APACHE II and SOFA scores as well as survival outcomes. Overall, the study addresses an important clinical challenge and provides relevant findings, but certain methodological and interpretative issues require clarification.
Response: We would like to sincerely thank the reviewers for their careful evaluation of our manuscript and for the many constructive and insightful comments provided. We greatly appreciate the time and expertise invested in this review process. The suggestions and remarks have been extremely helpful in improving the clarity, scientific rigor, and overall quality of our work. We have carefully revised the manuscript in response to each point raised and believe that these changes have strengthened the study. Our detailed responses to individual comments are provided below.
- Title is very long. Make it short and informative.
Response:
We appreciate this suggestion. In accordance with the reviewer’s recommendation, we have shortened the title to make it more concise while preserving its informative value.
The revised title is: “sTREM-1, HMGB1, CRP, PCT, sCD14-ST, IL-6, IL-10, sHLA-G, and vitamin D in Relation to Clinical Scores and Survival in SIRS/Sepsis”. This new version (see title page, line 1) maintains the focus on key biomarkers and their relationship with clinical outcomes, while being shorter and easier to follow.
- The topic is timely and clinically relevant. The role of inflammatory biomarkers in improving early diagnosis and prognosis of sepsis remains an important unmet need, and the manuscript contributes valuable insights, particularly regarding sTREM-1, IL-10, and vitamin D.
Response:
We sincerely thank the reviewer for this positive assessment of our work and for recognizing the clinical relevance of our study. Our aim was to highlight the diagnostic and prognostic potential of selected biomarkers in systemic inflammatory response and sepsis, with particular emphasis on sTREM-1, IL-10, and vitamin D. We are pleased that the reviewer acknowledges the value of these findings. No changes to the manuscript were necessary in this regard.
- The abstract is comprehensive, but overly long and dense. It would benefit from greater conciseness with a clearer distinction between diagnostic and prognostic findings.
Response:
We thank the reviewer for this valuable suggestion. In response, we have substantially revised the abstract to improve clarity and conciseness. The revised version is shorter and now clearly distinguishes diagnostic markers (sTREM-1, CRP, PCT, sCD14-ST, and neutrophil-to-lymphocyte ratio) from prognostic markers (IL-10 and vitamin D). These modifications can be found in the revised Abstract (section “Abstract”).
- The introduction provides good background on sepsis immunopathogenesis (SIRS, CARS, PICS), but some sections are lengthy and repetitive. Condensing this part would enhance readability without losing scientific depth.
Response:
We thank the reviewer for this helpful observation. We have carefully revised the Introduction (section “1. Introduction”) to improve readability by shortening and condensing repetitive passages. Specifically, we streamlined the background description of sepsis immunopathogenesis (SIRS, CARS, and PICS) and removed overlapping content on immune dysregulation. While keeping the essential references and scientific context, we reduced redundancy to provide a clearer and more concise background.
- The rationale for including such a broad panel of biomarkers is well explained; however, the novelty compared with prior studies should be emphasized more clearly. Some cited literature overlaps with the current findings, so highlighting the distinct contribution of this cohort would strengthen the justification.
Response:
We thank the reviewer for this important remark. In response, we have revised the Introduction (end of section “1. Introduction”) to better highlight the novelty of our work. While previous studies have examined some of these biomarkers individually, our study is distinct in its simultaneous assessment of a broader panel—including sTREM-1, HMGB1, sCD14-ST, IL-10, vitamin D, and sHLA-G—in the same cohort of SIRS and sepsis patients. This approach allowed us to directly compare their diagnostic and prognostic potential within one study setting. In particular, we provide additional evidence for the association of vitamin D and IL-10 with short- and intermediate-term survival, as well as their correlations with APACHE II and SOFA scores. By evaluating these markers together in a well-defined patient population, our work contributes complementary insights to the existing literature.
- The study population is relatively small (n=43) and heterogeneous. This limits statistical power and generalizability. The authors should acknowledge this limitation more explicitly and avoid over-interpretation.
Response:
We fully agree with the reviewer’s observation. We have now explicitly addressed this limitation in the Discussion (section “4. Discussion”), where we note that the relatively small and heterogeneous study cohort (n=43) reduces the statistical power and limits the generalizability of our findings. We have also revised the wording of our conclusions to avoid over-interpretation and to emphasize that the results should be validated in larger, prospective cohorts.
- Inclusion and exclusion criteria are appropriate, but further detail on patient comorbidities and antimicrobial therapy would be useful since these factors may influence biomarker levels.
Response:
We thank the reviewer for this valuable suggestion. In response, we have expanded the description of patient characteristics in the section 2. Subjects and Methods (section 2.1. Subjects and Sample Collection”). Specifically, comorbidities and baseline disease severity (APACHE II and SOFA scores) are now summarized in Table 1 (“Characteristics of the investigated individuals”). In addition, we clarified that all patients received antimicrobial therapy according to current clinical recommendations, with treatment tailored based on microbiological findings.
- The methods are generally sound, with clear descriptions of assays (ELISA, flow cytometry, routine labs). However, variability between assay kits from different manufacturers may affect comparability. Some comment on inter-assay reproducibility would be helpful.
Response:
We thank the reviewer for this important observation. For each biomarker, measurements were performed using the same lot of commercial ELISA kits from a single manufacturer, which ensured consistency within each assay. However, since different biomarkers were analyzed with kits from different manufacturers, we acknowledge that inter-assay variability between markers cannot be fully excluded. Flow cytometry analyses followed standardized procedures with routine calibration, and routine laboratory tests were performed in the hospital’s accredited clinical laboratory.
- The choice of statistical tests is appropriate, but the results section contains a large number of comparisons. The risk of type I error should be discussed, and adjustments for multiple testing (e.g., Bonferroni or FDR correction) should be considered.
Response:
We appreciate the reviewer’s observation. To address the concern regarding multiple comparisons, we applied the Benjamini–Hochberg false discovery rate (FDR) correction. Adjusted p-values are now reported in the section 4. Results, and the description of statistical methods has been updated in the section 3. Statistics accordingly.
- The finding that IL-10 correlates with worse outcomes is interesting but requires caution—IL-10 kinetics may vary during sepsis progression. The authors should discuss the potential for serial measurements and dynamic changes rather than single time-point values.
Response:
Thank you for your insightful comment regarding IL-10 kinetics. While our primary analyses are focused on values at ICU admission—based on the hypothesis that patients with poorer prognosis would present with elevated levels, we recognize that serial measurements represent a distinct and important aspect of biomarker behavior. To address this, we also performed a time-course evaluation and observed that IL-10 levels tended to decline over time, particularly in non-survivors. This pattern may reflect immune exhaustion or dysregulation during sepsis progression.
Due to the limited number of surviving patients at later time points, formal statistical testing of IL-10 kinetics was not feasible beyond T3 (at 7th day). However, a descriptive analysis revealed a consistent downward trend in IL-10 levels among non-survivors, suggesting a potential link between immune exhaustion and poor outcome.
- The role of HMGB1 is addressed, but its significance was limited in this cohort. The conclusion that HMGB1 might serve as a late-phase prognostic marker is reasonable but speculative, and this should be toned down.
Response:
We thank the reviewer for this valuable comment. We agree that the role of HMGB1 in our cohort was limited and that strong conclusions cannot be drawn. In the revised version of the manuscript, we have moderated our interpretation: HMGB1 results are now presented only in Table 7, and we have removed speculative statements about its prognostic potential from the text. We now emphasize that, while HMGB1 may be involved in later phases of systemic inflammation, its significance remains uncertain and requires validation in larger, prospective studies.
- The results regarding TREM-1/TREM-2 cell expression are inconclusive due to low sample size. While the authors acknowledge this, it should be stressed that further validation is necessary.
Response:
We thank the reviewer for this comment. In response, and in line with the reviewer’s concern, we have omitted the results regarding TREM-1 and TREM-2 cell expression from both the Results and Methods sections. We fully agree that further validation in larger patient cohorts will be necessary to clarify the potential role of these markers.
- The emphasis on vitamin D deficiency as a risk factor and prognostic indicator is compelling and supported by correlations. However, causality cannot be established, and supplementation trials would be needed to confirm therapeutic implications.
Response:
We thank the reviewer for this thoughtful comment. We fully agree that, although our results demonstrate significant associations between vitamin D levels and survival, causality cannot be inferred from this observational study. We have clarified this point in the Discussion (section “4. Discussion”), noting that our findings should be interpreted as correlative and hypothesis-generating. We also added that randomized supplementation trials would be necessary to establish whether vitamin D has a therapeutic effect in sepsis.
- Figures and tables are informative but numerous. Consolidation of some data presentations (e.g., combining Tables 2–4 or summarizing correlations graphically) would improve clarity.
Response:
We thank the reviewer for this helpful suggestion. We carefully considered the possibility of consolidating tables and converting some of the data into graphical form. However, we decided to retain the current structure, as each table presents distinct sets of results that would be difficult to merge without losing important details. We believe that keeping the tables separate provides greater transparency and allows readers to follow the analyses more clearly. To facilitate orientation, we have ensured that all tables and figures are clearly labeled and cross-referenced in the text.
- The English language is overall understandable but requires editing for grammar, style, and conciseness. Certain sentences are long and could be simplified to improve readability.
Response:
We thank the reviewer for this valuable comment. In response, we carefully revised the manuscript for grammar, style, and conciseness. Long and complex sentences were shortened and simplified to improve clarity and readability.
- The conclusion is consistent with the results, but stronger caution is needed regarding overstatement of clinical applicability. The biomarkers identified (particularly sTREM-1, IL-10, vitamin D) are promising but not yet validated for routine practice.
Response:
We thank the reviewer for this important remark. In response, we have revised the Conclusion (section “5. Conclusions”) to more clearly acknowledge the exploratory nature of our findings and to avoid overstating clinical applicability. The revised version highlights CRP, PCT, and sCD14-ST as reliable diagnostic markers, while underscoring the complementary prognostic value of IL-10 and vitamin D. Importantly, we now explicitly state that although these biomarkers are promising, they are not yet validated for routine clinical use and require confirmation in larger, multicenter cohorts.
- The references are adequate and up-to-date, but some could be shortened by focusing on the most relevant literature.
Response:
We thank the reviewer for this helpful remark. During the revision of the Introduction, we omitted some references to reduce redundancy and focus on the most relevant literature. At the same time, we added more recent citations to strengthen the scientific context. We believe the revised reference list now provides a balanced and up-to-date overview of the field.
Reviewer 2 Report
Comments and Suggestions for Authors
The manuscript submitted to the Biomedicines and entitled «ILevel of systemic inflammatory biomarkers sTREM-1, HMGB1, CRP, PCT, sCD14-ST, IL-6, and IL-10, vitamin D, HLA-G, CD64 expression, and other parameters, and their correlation with APACHE II and SOFA scores, and 7- and 28-day of survival in SIRS/sepsis patients» is aimed to access the plasma concentrations of sTREM-1, CRP, PCT, sCD14-ST, CD64 expression on neutrophils and the neutrophil-to-lymphocyte ratio as perspective indicators of systemic inflammation of infectious etiology. The presented retrospective case-control study is well-designed and performed on the high methodological level but there are some issues that must be clarified by the authors before publication.
1. The introduction section is too long and overloaded with the information. The introduction should be short and concise enough, briefly introduce the reader to the essence of the problem being studied and highlight its novelty and relevance. The authors need to revise this section and move some of the information to the Discussion section.
2. Authors must present full clinical and anamnestic data of patients not limited to data on gender and age described in the Table 1.
3. The protocol of laboratory analysis, chemistry, kits and equipments used for cellular TREM-1, TREM-2 expressions and plasma concentration of 25-hydroxyvitamin D (25(OH)D) must be briefly presented in the subsections 2.2.2. and 2.2.3., respectively. If the authors used standard protocols supplied with the commercial kits, relevant information should also be provided.
4. Did the authors conduct appropriate statistical analyses to control for the influence of gender, age, and other covariates on the results?
5. Authors must clearly state the novelty of their research. Serum blood levels of sTREM-1, CRP, PCT sCD14-ST, HMGB1 and VD in SIRS patients as well as its correlation with APACHE II and Sofa scores are actively studying for the last decades, a lot of articles on this topic are published. Authors should clarify what their research brings to modern biomedical science.
6. Given the above, the authors need to use more recent literary sources in their manuscript. In its current form, the manuscript contains only 5 sources less than 5 years old, the vast majority of the sources used are older than 10 years, which is very strange given the number of articles published in this topic.
7. I recommend that the authors conduct a ROC analysis that would show the predictive value of the studied biological markers and increase the significance of the obtained results.
8. Authors must format references in the text in accordance with the journal's requirements.
Author Response
Reviewer 2: The manuscript submitted to the Biomedicines and entitled «ILevel of systemic inflammatory biomarkers sTREM-1, HMGB1, CRP, PCT, sCD14-ST, IL-6, and IL-10, vitamin D, HLA-G, CD64 expression, and other parameters, and their correlation with APACHE II and SOFA scores, and 7- and 28-day of survival in SIRS/sepsis patients» is aimed to access the plasma concentrations of sTREM-1, CRP, PCT, sCD14-ST, CD64 expression on neutrophils and the neutrophil-to-lymphocyte ratio as perspective indicators of systemic inflammation of infectious etiology. The presented retrospective case-control study is well-designed and performed on the high methodological level but there are some issues that must be clarified by the authors before publication.
Response: We would like to sincerely thank the reviewers for their careful evaluation of our manuscript and for the many constructive and insightful comments provided. We greatly appreciate the time and expertise invested in this review process. The suggestions and remarks have been extremely helpful in improving the clarity, scientific rigor, and overall quality of our work. We have carefully revised the manuscript in response to each point raised and believe that these changes have strengthened the study. Our detailed responses to individual comments are provided below.
- The introduction section is too long and overloaded with the information. The introduction should be short and concise enough, briefly introduce the reader to the essence of the problem being studied and highlight its novelty and relevance. The authors need to revise this section and move some of the information to the Discussion section.
Response:
We thank the reviewer for this helpful advice. In response, we have revised and shortened the Introduction, focusing on the essential background needed to frame the study and emphasizing its novelty and relevance. Redundant or highly detailed information has been removed or, where appropriate, moved to the Discussion section. We believe this restructuring has improved clarity and readability while preserving the scientific context.
- Authors must present full clinical and anamnestic data of patients not limited to data on gender and age described in the Table 1.
Response:
We thank the reviewer for this comment. In response, we have expanded Table 1 (“Characteristics of the investigated individuals”) to include additional clinical and anamnestic information, specifically the most frequent comorbidities of the patients, along with baseline disease severity (APACHE II and SOFA scores). We believe these additions provide a more complete overview of the study cohort.
- The protocol of laboratory analysis, chemistry, kits and equipments used for cellular TREM-1, TREM-2 expressions and plasma concentration of 25-hydroxyvitamin D (25(OH)D) must be briefly presented in the subsections 2.2.2. and 2.2.3., respectively. If the authors used standard protocols supplied with the commercial kits, relevant information should also be provided.
Response:
We thank the reviewer for this comment. As one of the reviewers recommended omitting the results concerning TREM-1 and TREM-2 expressions due to the small number of tested patients, these analyses were removed from both the Results and the Methods sections. Regarding 25(OH)D, plasma concentrations were determined using an electrochemiluminescence binding assay (Elecsys Vitamin D total, Cobas; Roche Diagnostics GmbH, Mannheim, Germany) on an immunochemical analyzer at Laboratoria Piešťany, Ltd., Piešťany, Slovakia. This information has been added to the Methods (section 2.2.2.).
- Did the authors conduct appropriate statistical analyses to control for the influence of gender, age, and other covariates on the results?
Response:
We thank the reviewer for this insightful comment. To address this point, we performed multivariable logistic regression analyses for those biomarkers that showed statistically significant differences between SIRS and sepsis. In each model, we adjusted for age, sex, BMI, and relevant comorbidities. The adjusted results have now been incorporated into the revised Results section of the manuscript.
- Authors must clearly state the novelty of their research. Serum blood levels of sTREM-1, CRP, PCT sCD14-ST, HMGB1 and VD in SIRS patients as well as its correlation with APACHE II and Sofa scores are actively studying for the last decades, a lot of articles on this topic are published. Authors should clarify what their research brings to modern biomedical science.
Response:
We thank the reviewer for this important remark. We agree that many biomarkers, including sTREM-1, CRP, PCT, sCD14-ST, HMGB1, and vitamin D, have been previously investigated in the context of sepsis and SIRS. To address this, we have revised the Introduction (section “1. Introduction”) to more clearly state the novelty of our study. Unlike most prior reports, we simultaneously analyzed a broad panel of inflammatory and regulatory biomarkers—including IL-10 and sHLA-G—within the same cohort of SIRS and sepsis patients, which allowed us to directly compare their diagnostic and prognostic potential. Importantly, our study highlights the combined role of IL-10 and vitamin D as predictors of survival and their correlations with established clinical scores (APACHE II and SOFA), providing additional insights that complement the existing literature. We believe this integrated approach adds value to the field of biomedical science by emphasizing how multiple biomarkers interact in systemic inflammatory conditions.
- Given the above, the authors need to use more recent literary sources in their manuscript. In its current form, the manuscript contains only 5 sources less than 5 years old, the vast majority of the sources used are older than 10 years, which is very strange given the number of articles published in this topic.
Response:
We thank the reviewer for this important advice. In response, we have carefully revised the reference list and increased the number of recent citations published within the last 5 years. We believe that the inclusion of these more up-to-date sources strengthens the scientific context and relevance of the manuscript.
- I recommend that the authors conduct a ROC analysis that would show the predictive value of the studied biological markers and increase the significance of the obtained results.
Response:
We thank the reviewer for this valuable suggestion. In response, we performed additional ROC analyses to evaluate the predictive value of the studied biomarkers. The new ROC curves and corresponding AUC values are now presented in the Results (section “3. Results”) and illustrated in Figure 1,2 and 3. We believe these analyses strengthen the robustness and clinical relevance of our findings.
- Authors must format references in the text in accordance with the journal's requirements.
Response:
We thank the reviewer for this comment. All references in the text and in the reference list have now been reformatted to fully comply with the journal’s style requirements.
Reviewer 3 Report
Comments and Suggestions for Authors
The authors focused on the correlation between inflammatory markers of sepsis and disease progression by studying a novel biomarker combination for systemic inflammation (SIRS/sepsis) including sTREM-1, HMGB1, and vitamin D, addressing the shortcomings of traditional diagnostic markers like CRP and PCT in sensitivity and specificity. Special attention was given to the association of sTREM-1 with clinical scoring systems (APACHE II, SOFA), providing new insights for early diagnosis and prognostic evaluation. The specific suggestions are as follows.
- The grouping is clear (SIRS, sepsis, septic shock), but it is not specified whether randomization or matching of baseline characteristics (such as age, comorbidities) was performed.
- The sample size is relatively small (43 cases). Although biases were controlled through strict inclusion/exclusion criteria, this may affect statistical power. It is recommended to expand the sample size in the future to validate the conclusions.
- Table 2-5 is clear, it is suggested to supplement the survival analysis with Kaplan-Meier curves.
- It is not mentioned whether the experimental operation used a blind method.
- The mechanisms that have not been fully explored, such as whether septic-related inflammatory factors operate through the TLR signaling pathway or the STING-related signaling pathway(Ref. PMID: 39939798 DOI: 10.1038/s41418-025-01457-z ). It is recommended to elaborate on this in the discussion or introduction section.
- Does Vitamin D regulate inflammation through the NF-κB pathway?
- It is recommended to add specific threshold ranges for biomarker standardization or to discuss them.
Author Response
Reviewer 3: The authors focused on the correlation between inflammatory markers of sepsis and disease progression by studying a novel biomarker combination for systemic inflammation (SIRS/sepsis) including sTREM-1, HMGB1, and vitamin D, addressing the shortcomings of traditional diagnostic markers like CRP and PCT in sensitivity and specificity. Special attention was given to the association of sTREM-1 with clinical scoring systems (APACHE II, SOFA), providing new insights for early diagnosis and prognostic evaluation. The specific suggestions are as follows.
Response: We would like to sincerely thank the reviewers for their careful evaluation of our manuscript and for the many constructive and insightful comments provided. We greatly appreciate the time and expertise invested in this review process. The suggestions and remarks have been extremely helpful in improving the clarity, scientific rigor, and overall quality of our work. We have carefully revised the manuscript in response to each point raised and believe that these changes have strengthened the study. Our detailed responses to individual comments are provided below.
- The grouping is clear (SIRS, sepsis, septic shock), but it is not specified whether randomization or matching of baseline characteristics (such as age, comorbidities) was performed.
Response:
We thank the reviewer for this important comment. Randomization or matching of baseline characteristics was not performed, as this was an observational study based on consecutive ICU admissions. To address potential imbalances, we performed multivariable logistic regression analyses specifically for those biomarkers that showed statistically significant differences between SIRS and sepsis. In each model, we adjusted for age, sex, BMI, and relevant comorbidities. The adjusted results have been incorporated into the revised Results section of the manuscript.
- The sample size is relatively small (43 cases). Although biases were controlled through strict inclusion/exclusion criteria, this may affect statistical power. It is recommended to expand the sample size in the future to validate the conclusions.
Response:
We thank the reviewer for this valuable observation. We fully agree that the relatively small cohort size (n=43) limits statistical power and the generalizability of our findings. To minimize bias, we applied strict inclusion and exclusion criteria. We emphasize that our results should be interpreted as exploratory and require confirmation in larger, prospective multicenter cohorts. We plan to further enlarge the study cohort. Expanding the sample size will be essential to validate our conclusions in the future.
- Table 2-5 is clear, it is suggested to supplement the survival analysis with Kaplan-Meier curves.
Response:
We thank the reviewer for this helpful suggestion. In response, we have added Kaplan–Meier survival curves. Specifically, we included a survival plot for IL-10 stratified by serum levels (<900 vs. ≥900 pg/mL), which demonstrated a significant difference confirmed by both log-rank and Wilcoxon tests (p < 0.0001). We also evaluated vitamin D, where the survival curve showed a trend toward improved outcomes in patients with higher levels, although the difference did not reach statistical significance. These visual results complement our Cox regression analysis, which confirmed IL-10 as a significant predictor and vitamin D as borderline. The corresponding figures and description have been added to section 3. Results.
- It is not mentioned whether the experimental operation used a blind method.
Response:
We thank the reviewer for this important comment. We clarify that the laboratory personnel who performed the biomarker analyses did not have access to patients’ clinical data or information about their allocation to subgroups (SIRS vs. sepsis). Thus, laboratory measurements were carried out independently of clinical classification. However, complete blinding of the study was not feasible, and we acknowledge this as a methodological limitation. We have now explicitly stated this in the Methods (section “2.1. Subjects and Sample Collection”).
- The mechanisms that have not been fully explored, such as whether septic-related inflammatory factors operate through the TLR signaling pathway or the STING-related signaling pathway (Ref. PMID: 39939798 DOI: 10.1038/s41418-025-01457-z). It is recommended to elaborate on this in the discussion or introduction section.
Response:
We thank the reviewer for this valuable suggestion. In response, we have elaborated on the potential involvement of TLR and STING signaling pathways in the Introduction. We added the following text:
“TREM-1 work together with Toll-Like Receptors (TLRs) to amplify inflammatory responses. TLRs recognize various microbial and endogenous ligands, and signaling then upregulates TREM-1 expression. After TLRs triggers an initial inflammatory cascade, activation of TREM-1 initiates results in the production of pro-inflammatory cytokines and chemokines, and functions as an amplifier of inflammation. Although, TREM-1 was initially predominantly associated with infectious diseases (first in response to bacterial lipopolysacharide), it plays a great role also in sterile inflammatory diseases (Baykara et al., 2024; Colonna, 2023; Li et al., 2024; Liu et al, 2018; Panagopoulos et al., 2022).
During infectious and non-infectious inflammation besides TLR and TREM-1 pathways, so called STING (stimulator of interferon genes), pathway can be triggered. This pathway is activated by exogenous DNA, such as bacterial DNA, viral DNA and other exogenous DNA fragments, endogenous DNA, such as cytosol self-DNA and cytoplasmic chromatin fragment leakage. Many nucleic acid sensors have been identified to detect pathogen and cellular damage. It remains unclear whether STING is activated independently by septic stimuli or as a downstream effect of TLR engagement (Sun et al., 2025; Zhou et al., 2020).”
- Does Vitamin D regulate inflammation through the NF-κB pathway?
Response:
We thank the reviewer for this interesting question. Yes, vitamin D does regulate inflammation through the NF-κB pathway. According to recent research, vitamin D—specifically its active form, calcitriol (1,25-dihydroxyvitamin D)—binds to vitamin D receptors (VDRs), which are present in many immune cells including T cells, B cells, macrophages, and dendritic cells. This binding initiates a cascade of gene regulation that includes:
- Inhibition of the NF-κB pathway: Vitamin D suppresses the activation of NF-κB, a key transcription factor that promotes the expression of pro-inflammatory genes. By doing so, it reduces the production of cytokines such as TNF-α, IL-6, and IL-1β.
- Promotion of anti-inflammatory cytokines: It enhances the expression of cytokines like IL-10, contributing to a more regulated and less aggressive immune response.
- Modulation of immune cell differentiation: Vitamin D influences the maturation and function of dendritic cells and macrophages, steering them toward a more tolerogenic (anti-inflammatory) phenotype.
These mechanisms underscore vitamin D’s potential as a therapeutic agent in inflammatory and autoimmune diseases, including rheumatoid arthritis, inflammatory bowel disease, and multiple sclerosis.
- It is recommended to add specific threshold ranges for biomarker standardization or to discuss them.
Response:
We appreciate the reviewer’s suggestion to define specific threshold ranges for biomarker standardization. However, due to the exploratory nature and limited sample size of our study, we do not consider it appropriate to propose clinically applicable cut-off values at this stage. In our analysis, stratification was based on distribution within the cohort and was intended to highlight potential prognostic trends rather than establish definitive thresholds. We acknowledge that future studies with larger populations and external validation will be required to determine standardized and clinically meaningful cut-off values.
Round 2
Reviewer 2 Report
Comments and Suggestions for Authors
The authors corrected all issues and improved their manuscript. It can be published in the current form.
Reviewer 3 Report
Comments and Suggestions for Authors
Accept for publication.